# Low Detection Limit and High Sensitivity 2-Butanone Gas Sensor Based on ZnO Nanosheets Decorated by Co Nanoparticles Derived from ZIF-67

**DOI:** 10.3390/nano13172398

**Published:** 2023-08-23

**Authors:** Hua Zhang, Wenjie Zhao, Fanli Meng

**Affiliations:** College of Information Science and Engineering, Northeastern University, Shenyang 110819, China; 2170901@stu.neu.edu.cn

**Keywords:** 2-Butanone detection, nanosheet, cobalt-modified ZnO, heterojunction

## Abstract

2-butanone has been certified to cause potential harm to the human body, environment, etc. Therefore, achieving a method for the high sensitivity and low limit detection of 2-butanone is of great significance. To achieve this goal, this article uses ZIF-67 prepared by a precipitation method as a cobalt source, and then prepares cobalt-modified zinc oxide nanosheets through a hydrothermal method. The microstructure of the materials was observed by SEM, EDS, TEM, HRTEM, XPS and XRD. The test data display that the sensor ZC2 can produce a high response (2540) to 100 ppm 2-butanone at 270 °C, which is 21 times higher than that of pure ZnO materials. Its detection limit is also optimized to 24 ppb. The sensor (ZC2) also excels in these properties: selectivity, repeatability and stability over 30 days. Further analysis indicates that the synergistic and catalytic effects of p-n heterojunction are the key sources for optimizing the performance of sensors for detecting 2-butanone.

## 1. Introduction

2-Butanone is widely used in various industries such as food processing, the coating industry, electronic manufacturing, pharmaceuticals and cosmetics [1,2,3,4]. When the human body is exposed to 2-butanone in the atmosphere, 2-butanone can cause serious negative effects on the human respiratory system, central system and other organs and systems [5,6,7]. At the same time, butanone as a volatile flammable and toxic compound not only damages the environment [8] but also is one of the raw materials for producing “crystal methamphetamine” [9], which may lead to illegal issues such as drug dissemination and can threaten social security and public order. Therefore, the preparation of a sensor for detecting 2-butanone with a high sensitivity and low LOD is of great significance for human health, natural preservation, public safety precautions, production safety, etc.

Semiconductor metal oxides (SMO) have always been a hot research topic in the field of detection due to their cheapness and excellent gas sensing performance [10]. Various SMOs, for instance ZnO [11], In_2_O_3_ [12], CuO [13], Fe_2_O_3_ [14], Co_3_O_4_ [15], SnO_2_ [16], have been universally studied in the field of volatile gas detection. Among them, ZnO (n-type SMO) has a strong electron mobility and bonding ability, and is superior due to its cheapness, non-toxic properties, uncomplicated preparation and superior gas sensing performance [17,18]. Therefore, ZnO has always been a popular material in the research of metal oxide semiconductors. Numerous studies have shown that doping, noble metal modification and the establishment of heterostructures can effectively improve the gas sensing characteristics of zinc oxide [19,20,21,22]. Among them, the construction of a heterojunction can accelerate electron transfer and expand the electron depletion region, thus improving the sensing performance [23]. As a type of heterostructure, the p-n junction has been proven by some scholars to be a key factor in improving the gas sensing performance of sensors. Sikai Zhao et al. [24] prepared a p-n structure of copper-modified zinc oxide nanowires through a two step method. At 300 °C, the sensor has a response of 28 to 100 ppm ethanol, which is four times that of pure zinc oxide. Zihao Song et al. [25] prepared a p-n structure of NiO particles loaded on zinc oxide; the most ideal operating temperature of the sensor is reduced by 100 °C compared to pure zinc oxide. Lianyun Cheng et al. [26] successfully constructed a p-n junction based on Bi_2_O_3_-ZnO, whose response speed decreased from 6 s to less than 1 s compared with pure ZnO. This method can lead to multiple performance improvements in SMO and provides a new idea for our research on butanone detection.

Cobalt oxide is a p-type SMO, which can build a p-n junction with various n-type semiconductor materials on the contact surface to improve the response of pure inductive materials. This performance improvement is due to the mutual catalysis between Co_3_O_4_ and n-type semiconductor materials [27]. However, the accumulation of cobalt nanoparticles usually occurs in actual composite synthesis, resulting in the degradation of gas sensitive properties [28]. In recent years, MOFs have been shown to be effective precursors and templates for the preparation of functional micro-nanomaterials, with many advantages such as layered pore distribution, high surface area, morphology and good dispersion at the nanoscale [29,30]. Therefore, cobalt nanoparticles derived from ZIF-67 may be more uniformly modified on the surface of zinc oxide and better perform the synergistic effect of the two. In the meanwhile, the composite of cobalt and various semiconductor materials has been reported in the literature [31,32]; the heterojunction of cobalt oxide and zinc oxide has not been applied in butanone gas sensors. Therefore, the synthesized cobalt-modified zinc oxide nanostructures have a scientific and practical significance.

In this paper, ZIF-67 was prepared as a cobalt source by a precipitation method and the heterostructures of cobalt-modified zinc oxide nanosheets were prepared. According to the analysis of the gas sensitive properties of the system, the cobalt-modified zinc oxide nanosheet sensor shows a higher sensitivity, lower optimal operating temperature, lower detection limit, good reproducibility and stability under the optimal modification ratio.

At the same time, the reasons for the enhancement of the performance of cobalt-modified zinc oxide nanosheet sensors were also discussed, which are related to the formation of heterojunctions and the high catalytic activity of cobalt nanoparticles.

## 2. Preparation and Characterization

### 2.1. Preparation of ZIF-67

5.5 g dimethylimidazole was added into 20 mL ultrapure water and a blender was used to stir at a speed of 500 r/min for 5 min to form a clarified solution named solution A. As in the previous step, 0.45 g cobalt nitrate hexahydrate was dissolved in 30 mL ultrapure water to form solution B. We poured A slowly into B along the drain and stirred at room temperature for 6 h. Then, the purple solid was obtained by washing with ethanol and ultrapure water 5 times alternately at a speed of 8000 r/min through the centrifuge. We dried the purple solids in an incubator at 60 °C overnight.

### 2.2. Preparation of Pure ZnO and Co@ZnO

First, 3 mmol zinc nitrate hexahydrate, 8 mmol NaOH and 2 mmol CHBrN were dissolved in 32 mL ultrapure water and stirred for 10 min. Then, the solution was ultrasonic for 10 min to completely dissolve the drug and then stirred for 20 min to obtain a uniform solution. Subsequently, the mixed liquid was decanted into a high-pressure reactor (50 mL) and kept at 150 °C for 10 h in an electric furnace. When the reaction kettle reaches a natural temperature, the precipitate in it was centrifuged 5 times with deionized water and ethanol to obtain the precursor. Finally, the solid was dried overnight in a 60 °C incubator and grinded into powder. We placed the powder in a tube furnace and it was calcined at a high temperature of 400 °C for 2 h.

We weighed 26 mg ZIF-67, dissolved in 5 mL deionized water, and then ultrasound for 10 min. Then, 100 μL, 200 μL and 400 μL ZIF-67 solutions were extracted and placed in the preparation process of ZnO, and other processes remained unchanged. In the following discussion, samples with 0 μL, 100 μL, 200 μL and 400 μL ZIF-67 solutions added to the sample preparation process are labeled ZC0, ZC1, ZC2, and ZC4, respectively.

### 2.3. Characterization

The samples were studied by powder X-ray diffraction (XRD), scanning electron microscopy (SEM), transmission electron microscope (TEM), X-ray Photoelectron Spectroscopy (XPS) and X-ray energy spectrometry (EDS).

The phase of the sample was measured by XRD (Cu Kα radiation, Shimazu XRD-7000, Japan, λ = 0.154 nm) from 10° to 80° (2θ) with a speed of 5°/min. The principle is that the diffraction phenomenon occurs when the X-ray is projected into the crystal under the influence of the atoms in the crystal. Different arrangements of atoms in different crystals result in different changes in the X-ray diffraction, so phase analysis can be carried out. The micromorphology of the materials was researched by SEM images obtained by SSX-550. The microstructure of the materials was researched by TEM (JEOL Ltd., Tokyo, Japan). Finally, the composition of Zn-Co-O was verified by EDS and XPS.

### 2.4. Assembly and Evaluation of Sensors

First, we weld the four-pin ceramic tube, resistance wire and sensor base to obtain the blank sensor. Then, the sample powder obtained through the above experimental steps is mixed with ethanol, and a uniform viscous colloid is obtained under the shock of the shaker. Finally, 8 μL colloids were dropped on the ceramic tube of the blank sensor using a pipette gun and the colloids were evenly distributed on the ceramic tube until the ethanol evaporated and a sensing layer of uniform thickness was formed.

Before testing the gas sensitivity of the sensor, we first placed the sensor on the aging table, aged the sensor at 180 °C for 36 h and then left it at room temperature for 72 h.

The gas sensitive performance test is carried out in a transparent gas chamber (PMMA) with a capacity of 1 L. The overall test system is shown in Figure 1. The hardware component includes a clear gas chamber, a computer, a tank of synthetic air (21% oxygen and 79% nitrogen), a source/measurement unit (SMU) and an adjustable regulated power source. We used Labview to build the software part. The SMU applies flat voltage (10 V) to the sensor and logs test data at a frequency of 2 Hz. By changing the output of the adjustable regulated power supply to change the temperature around the sensor, because the output of the power source directly acts on the resistance wire.

Prior to testing, the gas (21% oxygen and 79% nitrogen) is to be first injected into the closed air chamber for 10 min. This can remove the impure gas. Then, adjust the required working temperature and inject the required gas into the transparent chamber with injector after the resistance or current is stabilized. The test interface will display the change of the resistance or current in real time (the test system will automatically save the data). After the resistance or current changes are stabilized, synthetic air is passed to return the resistance or current to the initial value before the test.

The sensor designed in this paper calculates the sensitivity by the following equation:(1)$$S=\frac{R_{a}}{R_{g}}$$

In this equation, *R_g_* and *R_a_* represent the stabilized resistance of the sensor in the test gas and synthetic air, respectively.

## 3. Results

### 3.1. Structural Characteristics

In Figure 2, the XRD data of the ZC0, ZC1, ZC2 and ZC4 samples are matched with zinc oxide hexagonal Wurtzite (PDF#79-0206: ZnO). The diffraction peaks at 31.82°, 34.48°, 36.3°, 47.6°, 56.68°, 62.94°, 66.46°, 68.04°, 69.16°, 72.68° and 77.04° correspond to the (100), (002), (101), (102), (110), (103), (200), (112), (201), (004) and (202) planes of Wurtzite zinc oxide, respectively. Moreover, no diffraction peaks of other elements exist in the XRD spectra of the four samples, indicating that the four materials prepared had good crystallinity. At the same time, the diffraction peaks of the Co-related elements (CoO, Co_2_ O_3_ and Co_3_O_4_) were not detected, possibly because the doping amount of Co was too low to be detected by XRD [33]. For more characteristics of samples with different cobalt modification ratios, the XRD pattern of the (202) plane was selected for amplification. In Figure 2b, as growth of the cobalt content grows, the diffraction peak deviation angle of the sample increases, which indicates that part of the cobalt ions are mixed into the crystal lattice of zinc oxide. In this process, the main structure of zinc oxide remains unchanged but the crystal face spacing becomes smaller [34].

The Scherer formula [35] is cited in this article to compute the grain size:(2)$$d=\frac{k\times \lambda }{\beta \times Cos\theta }$$

In this formula, *K* represents a constant (value 0.89), *λ* represents the wavelength of the X-ray, Beta (*β*) is the width of the diffraction peak at the height of the half-peak, and θ is the diffraction angle of the X-ray. The seven strong peaks (100), (002), (101), (102), (110), (103) and (112) in the XRD spectrum are used to calculate the grain size (Table 1 is specific parameters). The grain size obtained by the XRD pattern is about 23.96 nm, 23.35 nm, 23.39 nm and 23.59 nm, respectively. After the analysis, it can be seen that the grain size of the sample changes slightly after the cobalt metamorphism treatment, but it is always smaller than that of pure zinc oxide and the difference is not large.

Figure 3a,b is the SEM images of pure zinc oxide. It can be seen that the material presents a monodisperse sheet microstructure, the surface of the nanosheet is smooth and flat, the length is less than 1 μm and the thickness is about 30 nm. Figure 3c–h shows the microstructure of ZC1, ZC2 and ZC4, respectively. It can be seen that the modification of cobalt does not affect the lamellar structure of the material, and the cobalt nanoparticles are almost uniformly attached to the surface of the zinc oxide nanoparticle without accumulation. With the modification of Co, the thickness of zinc oxide nanosheets is maintained at about 35 nm, which is about 5 nm thicker than ZC0, and its specific surface area will obviously be reduced. At the same time, with the modification of cobalt, the thickness of the cobalt-modified zinc oxide nanosheets decreases first and then increases. It can be seen from Figure 4a that the ZIF-67 consists of polytope with a dimension of 200–500 nm. Figure 4b shows the EDS spectral analysis of ZC2. It shows that the elements in ZC2 included zinc, cobalt and oxygen, and no other impurity elements were found. The mass proportions of O, Zn and Co in ZC2 were 51.73%, 47.76% and 0.51%, respectively, which provided evidence for the existence of Co.

Figure 5a–c shows the TEM and HRTEM images of the material (ZC2), respectively. From the TEM images it can be seen that cobalt nanoparticles are decorated on the surface of ZnO nanosheets. The lattice of the sample was analyzed by HRTEM images. The obtained results, when compared with standard cards, show that the lattice stripe of 0.26 nm corresponds to the (002) crystallographic plane of ZnO and the lattice stripe of 0.286 nm corresponds to the (220) crystallographic plane of Co_3_O_4_. In addition, the obvious stripe boundary between the two crystal planes proves the existence of p-n heterojunction [36].

In this paper, XPS was used to further demonstrate the coexistence of Zn and Co and the valence states they present. Figure 6a shows the measured XPS spectra of ZC0 and ZC2. The XPS spectrum of ZC2 shows that four elements, Zn, Co, O and C, are present in the sample and element C is the contamination caused by adsorption during the detection process. Figure 6b shows the precision spectrum of Zn 2p, there is a peak at 1044 eV and 1020.9 eV each, the energy difference is about 23.1 eV, indicating that Zn^2+^ is normally present in the material [37]. Figure 6c shows the precision spectrum of Co 2p, two peaks at 780.1 eV and 789.5 eV correspond to Co 2p3/2 and Co 2p1/2, respectively. They would be deconvoluted into two peaks. The peak corresponding to Co 2p_3/2_ consists of Co^3+^ (779.8 eV) and Co^2+^ (794.9 eV). The peak corresponding to Co 2p_1/2_ consists of Co^3+^ (794.61 eV) and Co^2+^ (797.26 eV). The difference in binding energy between the two peaks can be observed to be about 15.1 eV, revealing the presence of Co_3_O_4_ phase in the sample (ZC2) [38]. The atomic ratio of cobalt found by XPS analysis is 0.41%, which is similar to the result of the EDS measurement of 0.51%. Figure 6d shows the precision spectra of O 1 s showing contents of the lattice oxygen (O_L_), oxygen vacancy (O_V_) and adsorbed oxygen (O_C_) in ZC0 and ZC2. It can be found that the ratio of O_C_ increases from 12% to 14% for ZC2 compared to ZC0. And the ratio of O_V_ decreases from 23% to 18%, which may be due to the combination of cobalt atoms and oxygen vacancies.

### 3.2. Gas Sensing Characteristic

The operating temperature can change the electron mobility of the zinc oxide complex and provide active energy to break the reaction energy barrier for adsorption of the measured gas. Figure 7 shows the sensitivity of ZC0, ZC1, ZC2 and ZC4 at different operating temperatures. When the temperature rises to a critical point, the response value begins to decline; this critical point is the optimal operating temperature of the sensor. According to relevant studies, the reason for this phenomenon can be explained as follows: when the temperature of the sensor is lower than the critical point, it will lack sufficient active energy to promote the surface oxygen anion and 2-butanone REDOX reaction. Therefore, continuously increasing the temperature is a necessary means to find the optimal operating temperature that can provide the activation energy required for the complete reaction of the sensor. On the contrary, the desorption rate of 2-butanone will surpass the adsorption rate and its adsorption active site will also decrease rapidly, thus reducing the response [39]. The best operating temperature of ZC0 is 300 °C; as cobalt is decorated on the zinc oxide nanosheet, its best operating temperature begins to decrease. The best operating temperature of ZC1, ZC2 and ZC4 is 290 °C, 270 °C and 270 °C, respectively. ZC2 shows the best gas sensitivity with a response of 21.17 times that of ZC0 (120) at 270 °C (2540). The decrease of the optimum operating temperature can be owed to the decrease of the reaction energy barrier of adsorption of 2-butanone by the sensor due to the catalytic action of cobalt. The decrease in sensitivity (1749) of ZC4 may be due to excessive Co loading, which leads to the cobalt NPs occupying further active sites on the zinc oxide nanosheet, hindering electron transfer.

The sensitivity curves of ZC0, ZC1, ZC2 and ZC4 at different 2-butanone concentrations of 10–400 ppm at 300 °C, 290 °C, 270 °C and 270 °C, respectively, are shown in Figure 8a–d. It was found that as the concentration of 2-butanone increased, the sensitivity of all four sensors gradually increased. Under different concentration conditions, the sensitivity of ZC2 was higher than that of other sensors. Figure 7a,b shows the concentration gradient profiles of ZC0 and ZC2 in the detection of ppb level. It was found that the detection limit of ZC0 for 2-butanone is about 500 ppb, while that of ZC2 for 2-butanone is 24 ppb. It indicates that the modification of Co can optimize the LOD of zinc oxide on 2-butanone. At the same time, in Figure 9a,b, the baseline current of ZC2 is much smaller than that of ZC0. This is due to the strong oxygen adsorption capacity of Co nanoparticles, which can catalyze more oxygen ions to be adsorbed on zinc oxide nanosheets. It will greatly improve the primary resistance of zinc oxide and also provide the basic conditions for the improvement of sensor sensitivity [40].

The real-time variation curves of ZC0 and ZC2 are displayed in Figure 10. At 270 °C, 92 s and 30 s were the response time and recovery time of ZC2, respectively. Compared with ZC0 at 300 °C (58.25 s and 30.5 s), the performance slightly decreased. This may be because under the catalytic action of cobalt, ZC2 will adsorb more 2-butanone, which requires more time for its adsorption.

To verify the reproducibility of the sensors, the ZC0, ZC1, ZC2 and ZC4 sensors were measured for five cycles of 2-butanone gas at 100 ppm at 300 °C, 290 °C, 270 °C and 270 °C, respectively. Figure 11a–d shows the test results. The five cyclic response/recovery curves indicate that a single gas sensor has good repeatability.

To further analyze the selectivity of ZC0, ZC1, ZC2 and ZC4, the sensitivity of these 100 ppm measured gases (butanone, acetone, methanol, ethanol, ethyl acetate, formaldehyde, acetic anhydride, ammonia, butyl acetate, toluene and xylene) was evaluated by measuring the values. As shown in Figure 12, these sensors are all more responsive to 2-butanone than to other gases. Among them, ZC2 has a higher response to a variety of gases than other sensors and its response to butanone is 2.14 times higher than that of acetone (1211, the second highest response). In the above gas range, the selection ratio of ZC2 (sensitivity ratio of 2-butanone to other gases for comparison) can reach up to 423. The experimental outcomes attest that ZC0, ZC1, ZC2 and ZC4 exhibit good selectivity. The possible reason is that butanone molecules have more hydrogen atoms on the secondary carbon, which can quickly dissociate and react with oxygen ions [41].

Figure 13a shows the variation in the sensitivity of ZC2 and ZC0 under different humidity conditions. The sensitivity of both sensors decreases with increasing relative humidity, which may be caused by water molecules competing with 2-butanone for adsorption sites on the material surface [42]. In addition, the response of the ZC0 and ZC2 sensors to 100 ppm 2-butanone for 30 days was tested. The result is shown in Figure 13b. There is no significant error change in the sensitivity of ZC0 and ZC2, indicating that both have excellent stability.

In addition, Table 2 lists the research conducted by others on butanone gas sensors. It is found that the 2-butanone gas sensor (ZC2) designed in this paper has significant advantages in detection limit (24 ppb) and sensitivity (2540). Therefore, the cobalt-modified zinc oxide nanosheet sensor prepared in this article can provide a reference value for further research on 2-butanone detection.

### 3.3. Gas-Sensing Mechanism

The gas sensing mechanism can be explained through the principles of surface adsorption and electron transfer [44,45], and the sensor schematic diagram is shown in Figure 14. Pure zinc oxide stored in air attracts a large number of oxygen molecules in the air to adsorb on its surface; in the process, the oxygen molecules are transformed into adsorbed oxygen:(3)$$O_{2}(gas)\to O_{2}(ads)$$

Then, the electrons in the conduction band of zinc oxide are captured by *O*_2_(*ads*) and the *O*_2_(*ads*) become ions. The type of oxygen ion is determined by the temperature of its environment and the chemical equations for adsorption on the sensor surface at different operating temperatures are as follows [50]:(4)$$O_{2}(ads)+e^{-}\to {O_{2}}^{-}\;(T\leq 100\;{}^{\circ}C)$$



(5)
$${O_{2}}^{-}(ads)+e^{-}\to 2O^{-}\;(100\;{}^{\circ}C<T<300\;{}^{\circ}C)$$





(6)
$$O^{-}(ads)+e^{-}\to O^{2-}\;(T\geq 300\;{}^{\circ}C)$$



During the transformation of adsorbed oxygen into oxygen ions, a large number of electrons are extracted and bonded by adsorbed oxygen from the zinc oxide conduction band and an electron depletion layer is gradually formed on the surface of zinc oxide. As the butanone contacts the sensor, oxygen ions will REDOX with the butanone, producing water and carbon dioxide while releasing free electrons. With the return of free electrons, the electron depletion layer will gradually disappear and the resistance of the sensor will begin to decrease. At 270 °C, the reaction equation is as follows [51]:(7)$$C_{4}H_{8}O+{11O}^{-}\to 4CO_{2}+4H_{2}O+11e^{-}$$

The cobalt-modified zinc oxide is better than pure zinc oxide in terms of gas sensitive characteristics, which is rooted in three aspects:

First of all, cobalt oxide, as a p-type semiconductor, is easy to structure the p-n junction on the contact surface with zinc oxide (n-type semiconductor). The electrons of zinc oxide flow to cobalt oxide and the holes of cobalt oxide flow to zinc oxide and finally form the equilibrium of the Fermi level. In the air atmosphere, cobalt oxide will form a hole accumulation layer opposite the electron depletion layer of zinc oxide, which will greatly promote electron flow [23,32]. Therefore, an electron depletion layer wider than that of pure zinc oxide will form at the p-n junction. This will significantly enhance the primary resistance. This will greatly increase the initial resistance of the sensor. This large initial resistance value will greatly optimize the sensor’s earth-gas sensitivity. The REDOX reaction between 2-butanone and oxygen ions will release electrons and neutralize the holes at the same time [52]. This will effectively shrink the electron depletion layer and the hole accumulation layer and the resistance of the sensor will be sharply reduced, showing superior sensitivity. However, the continuous increase of cobalt content can lead to a decrease in sensitivity, which may be due to excessive cobalt NPs occupying a large number of active sites of zinc oxide.

In addition, the high catalytic and oxygen adsorption capacity of cobalt particles is another factor that optimizes the sensor’s enhanced gas sensitivity. The high catalytic performance of cobalt NPs effectively attenuates the energy barrier of the REDOX reaction between 2-butanone and oxygen ions. This makes the sensor more responsive and intensifies the attachment of oxygen ions to further thicken the electron depletion region [53,54]. This result accelerates the reaction of 2-butanone with abundant chemisorbed oxygen ions.

Finally, cobalt oxide particles derived from the MOF structure did not agglomerate during the modification process and were uniformly distributed on the surface of zinc oxide nanosheets. A larger surface area and more 2-butanone adsorption sites emanated from this.

Combined with the above factors, the cobalt-modified zinc oxide nanosheet sensor has an excellent gas sensitive performance in detecting butanone.

## 4. Conclusions

In this study, a zinc oxide nanosheet sensor (ZC2) modified with cobalt nanoparticles derived from ZIF67 was synthesized by a hydrothermal method using ZIF67 as a cobalt source. Gas sensitivity tests of this system show that for the detection of 100 ppm 2-butanone, the best working temperature of ZC2 drops to 270 °C and has a high response of 2540, which is 21 times higher than that of pure zinc oxide sensors. At the same time, the detection limit of ZC2 is reduced from 120 ppb to 24 ppb, which is an extremely low detection limit, and the sensor also has excellent repeatability and stability. The analysis shows that the synergistic and catalytic effects of p-n heterojunction have a decisive impact on optimizing the sensor’s performance in detecting 2-butanone. Meanwhile, the cobalt nanoparticles derived from ZIF67 inherited the dispersibility of ZIF-67 and were uniformly scattered with the ZnO nanosheets to maximize the catalytic function of cobalt nanoparticles.

## Figures and Tables

**Figure 1 nanomaterials-13-02398-f001:**
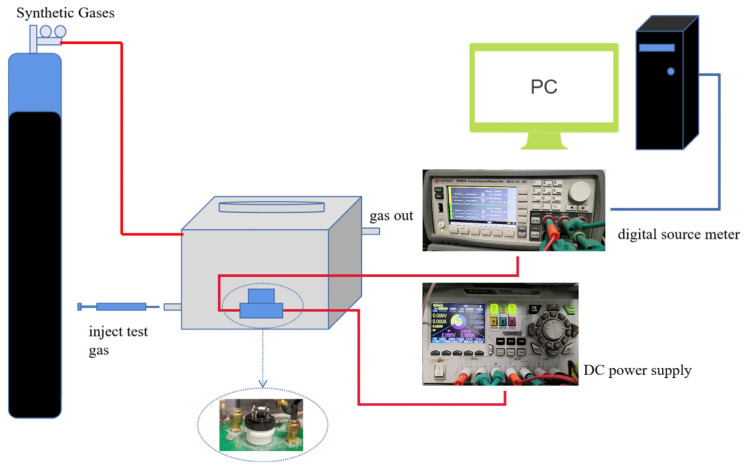
Measurement system of gas sensing property.

**Figure 2 nanomaterials-13-02398-f002:**
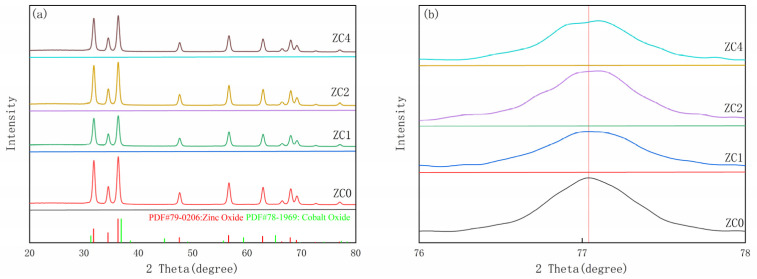
(**a**) XRD patterns of ZC0, ZC, ZC2 and ZC4. (**b**) The close spectrum of diffraction peaks between 76° and 78° in ZC0–ZC4.

**Figure 3 nanomaterials-13-02398-f003:**
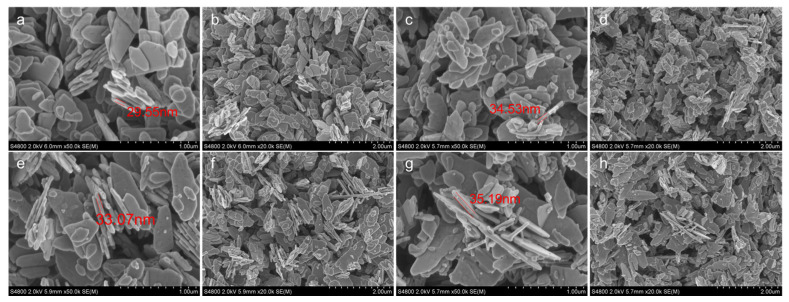
Scanning electron microscope pictures of ZnO nanosheets modified with increased amounts of cobalt in two microscopic sizes: (**a**,**b**) ZC0, (**c**,**d**) ZC1, (**e**,**f**) ZC2 and (**g**,**h**) ZC4.

**Figure 4 nanomaterials-13-02398-f004:**
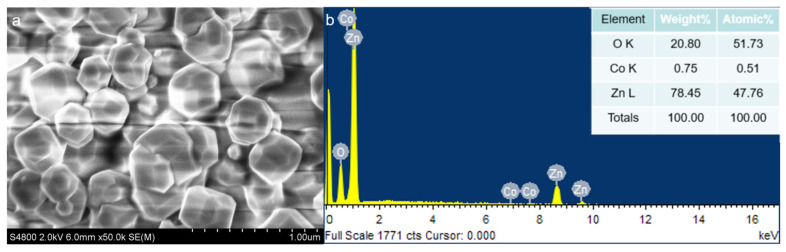
(**a**) Scanning electron microscope pictures of ZIF-67 and (**b**) EDS spectra of ZC2.

**Figure 5 nanomaterials-13-02398-f005:**
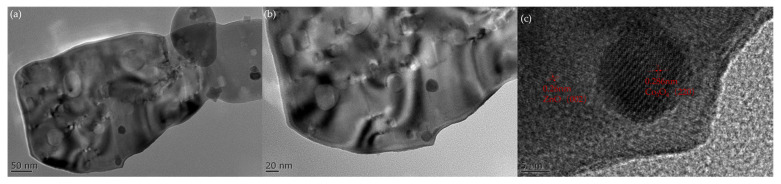
The TEM figures of ZC2 are (**a**,**b**); the HRTEM figures of ZC2 is (**c**).

**Figure 6 nanomaterials-13-02398-f006:**
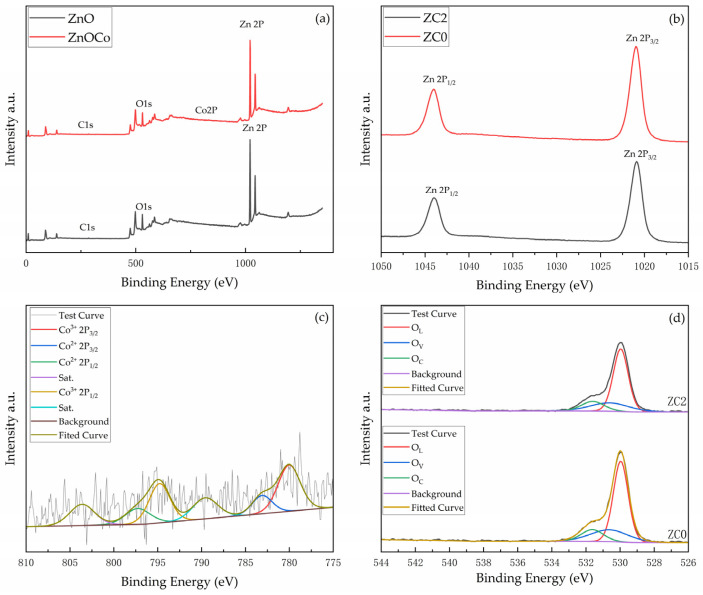
XPS spectra images of (**a**) the survey spectrum of ZC0 and ZC2, (**b**) Zn 2p spectrum of ZC0 and ZC2, (**c**) Co 2P spectrum of ZC2, (**d**) O 1s spectrum of ZC0 and ZC2.

**Figure 7 nanomaterials-13-02398-f007:**
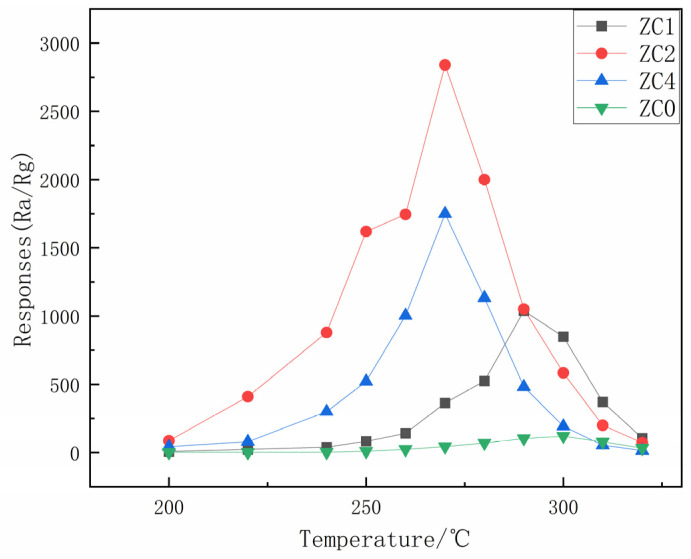
Responses of samples towards 100-ppm 2-butanone at varied operating temperature.

**Figure 8 nanomaterials-13-02398-f008:**
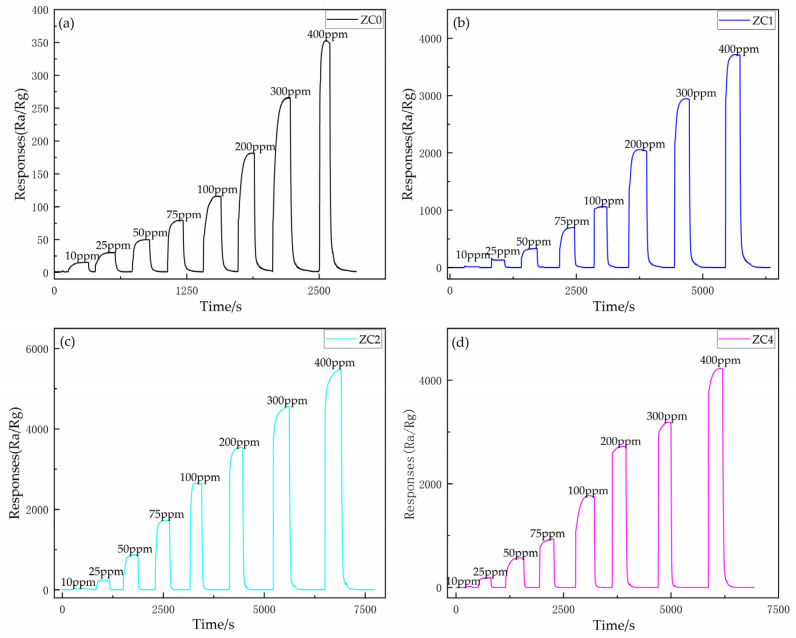
Static response curve of the sensor to a gradient viscosity (10–400 ppm) of 2-butanone at optimum operating temperature. (**a**) ZC0, (**b**) ZC1, (**c**) ZC2 and (**d**) ZC4.

**Figure 9 nanomaterials-13-02398-f009:**
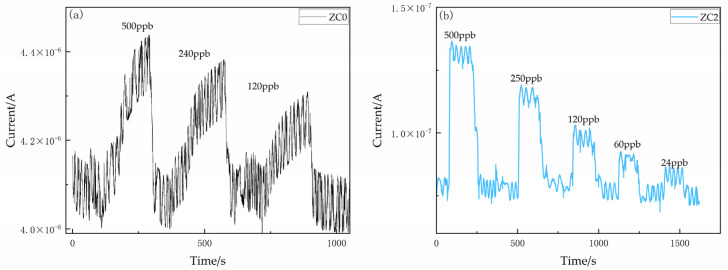
Static response curve of the sensor to a gradient density (ppb) of 2-butanone at top-notch operating temperature. (**a**) ZC0 and (**b**) ZC2.

**Figure 10 nanomaterials-13-02398-f010:**
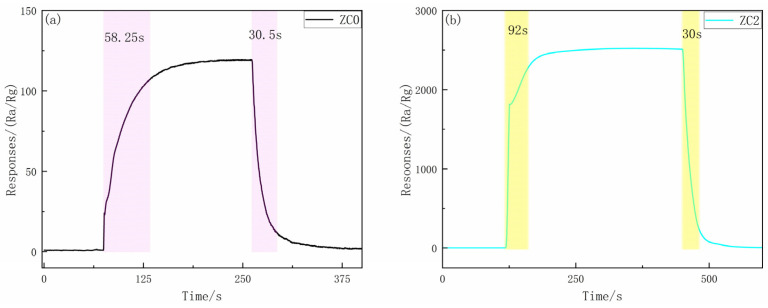
Transient current curve for sensors (**a**,**b**) exposed to 100 ppm 2-butanone at 300 °C and 270 °C, respectively.

**Figure 11 nanomaterials-13-02398-f011:**
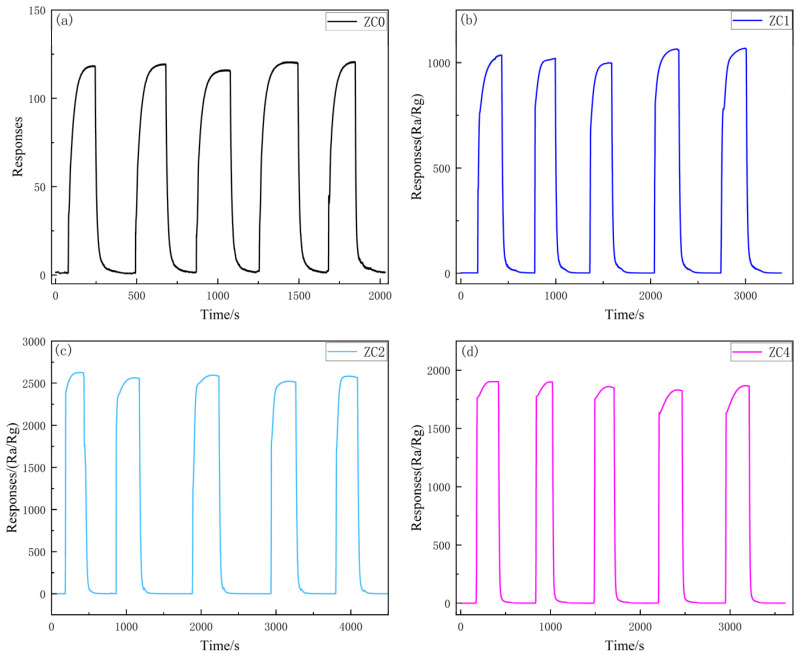
(**a**) Response profiles for five cycles of the sensor based on (**a**) ZC0, (**b**) ZC1, (**c**) ZC2 and (**d**) ZC4 to 100 ppm 2-butanone at 300 °C, 290 °C, 270 °C and 270 °C, respectively.

**Figure 12 nanomaterials-13-02398-f012:**
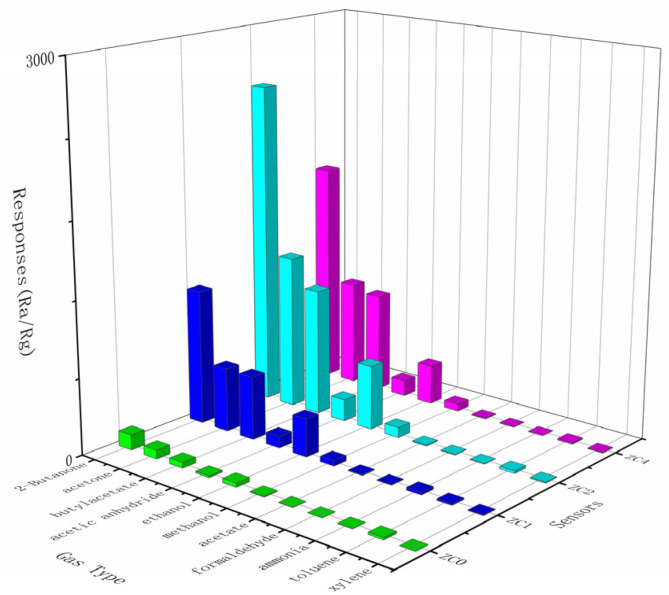
Selective analysis of sensors for 100 ppm different gases at their respective top-notch operating temperature.

**Figure 13 nanomaterials-13-02398-f013:**
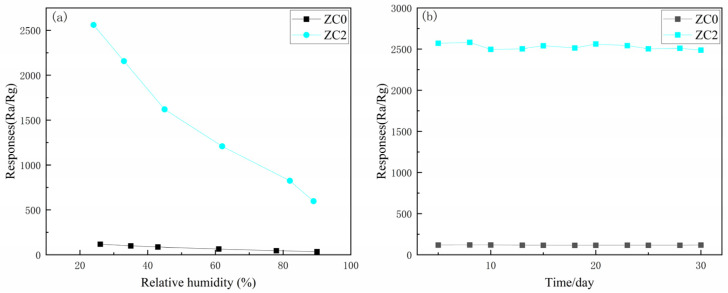
(**a**) Variation in sensitivity of ZC0 and ZC2 in the interval from 25% to 90% relative humidity; (**b**) 30 day stability data of 100 ppm 2-butanone from ZC0 and ZC2 sensors.

**Figure 14 nanomaterials-13-02398-f014:**
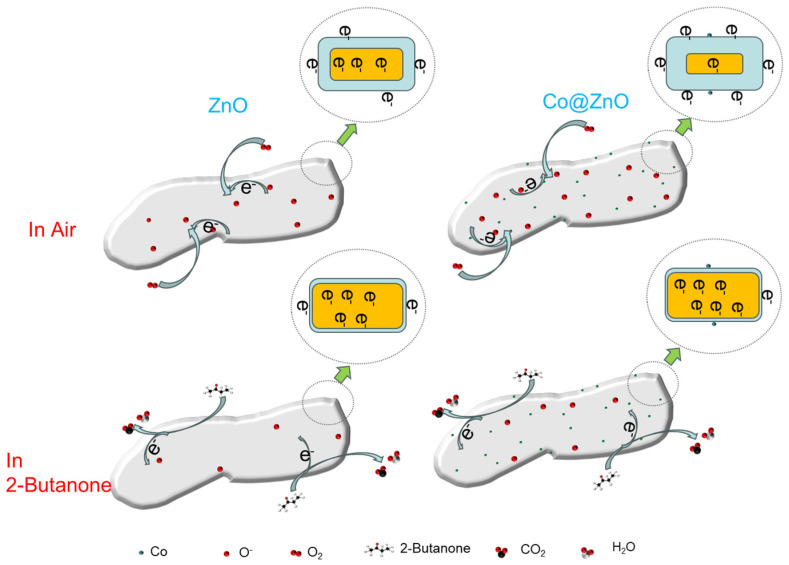
The gas adsorption sensing mechanism diagram.

**Table 1 nanomaterials-13-02398-t001:** Crystal parameters of ZC0, ZC1, ZC2 and ZC4.

Materials	Diffraction Planes (hkl)	Diffraction Angles (°)	FWHM (β)	Crystallite Size (nm)	Average Crystallite Size (nm)
ZC0	100	31.82	0.52953	17.47577235	23.95759925
	002	34.48	0.52478	18.17698595	
	101	36.3	0.54565	17.88077987	
	102	47.6	0.50504	23.08960689	
	110	56.68	0.55915	25.60044955	
	103	62.94	0.5546	31.16584861	
	112	68.04	0.61278	34.3137515	
ZC1	100	31.82	0.57065	16.21650001	23.3490385
	002	34.48	0.55618	17.15077616	
	101	36.3	0.56205	17.35903841	
	102	47.6	0.50995	22.86729104	
	110	56.66	0.56463	25.33853197	
	103	62.94	0.5613	30.79383509	
	112	68.04	0.62362	33.71729682	
ZC2	100	31.82	0.55351	16.71866043	23.37879741
	002	34.48	0.55274	17.25751472	
	101	36.3	0.56897	17.14791208	
	102	47.58	0.50976	22.8670741	
	110	56.66	0.566	25.27720019	
	103	62.94	0.56711	30.47835453	
	112	68.04	0.62017	33.90486583	
ZC4	100	31.82	0.55828	16.57581452	23.58591742
	002	34.46	0.55332	17.23529436	
	101	36.3	0.55863	17.46531252	
	102	47.6	0.51183	22.78329732	
	110	56.66	0.55973	25.56035108	
	103	62.92	0.55663	31.03098651	
	112	68.04	0.61035	34.4503656	

**Table 2 nanomaterials-13-02398-t002:** Comparison of sensing ability of 2-butanone sensors.

Material	T. (°C)	Conc. (ppm)	Lim. (ppm)	τ_res_./τ_rec_. (s)	Resp.	Ref.
ZnO/Pt twin-rods	450	100	5	8/-	35.2	[43]
SiO_2_@CoO	350	100	5	-/-	44.7	[44]
Ce/SnO_2_	175	20	0.5	20/-	23.9	[45]
Pd/SnO_2_	250	1000	-	1/35	451	[46]
0.5% APTES/ZnO	260	50	0.2	1/70	32	[47]
ZnSn_2_O_4_	300	100	0.1	6/47	80	[42]
bicone-like ZnO	400	100	0.41	-/-	29.4	[48]
NiS-H-ZnO	350	100	0.1	-/-	138	[49]
Co@ZnO nanosheet	270	100	0.024	92/30	2540	This work

## Data Availability

Not applicable.

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
