# Peer review of "Low Detection Limit and High Sensitivity 2-Butanone Gas Sensor Based on ZnO Nanosheets Decorated by Co Nanoparticles Derived from ZIF-67"

_nanomaterials, 2023, doi:10.3390/nano13172398_

Round 1
Reviewer 1 Report
The MS reports on the synthesis and development of [heterostructured] Co-ZnO based gas sensor for 2-butanone detection. The authors claimed that the as-developed gas sensor exhibits relatively high sensitivity with capability of detecting as low concentration of 24 ppb of butanone. The MS presents an interesting findings and can be published provided that the ff points are well addressed.
1. The authors stated that the as-fabricated nanostructured film is a heterostructure and the sensor's high sensitivity and low detection limit is attributed to the synergetic p-n effect of the hybrid nanostructure. However, the XRD data show that no cobalt oxide (CoO, Co2O3, Co3O4) related structure were detected. Hence, the sensor film was totally Co nanoparticle decorated ZnO nanosheets (Co-ZnO) and the nanoparticles can only act as a catalytic surfaces and enables the spill-over effect. Where is then the p-type SMO to form a p-n(ZnO) junction? It is only the ZnO which is acting as an n-type SMO.
2. The authors should further characterize the sensor film to investigate whether the as-synthesized film is truly a heterostructure with CoxOy - ZnO (p-n) as indicated in the MS (pp 12, Line 315-317).
3. The authors should present clear and readable graphs, for example, Fig.10 (a), it is impossible to read the name of gases in the plot.
4. There are many spelling and grammatical errors throughout the MS, [P-pattern SMO] on pp11 Line 294, and [Electron] in the middle of the sentence, etc.
Reviewer 2 Report
The manuscript entitled "Low detection limit and high sensitivity 2-butanone gas sensor based on ZnO nanosheets decorated by Co nanoparticles derived from ZIF-67" investigates a gas sensor for 2-butanone, fabricated using cobalt-modified ZnO nanosheets. The study demonstrates a notably low detection limit of 24 ppb. Overall, the main points and sensing performance are well-presented with appropriate experimental evidence. I believe that, with minor revisions, this manuscript can meet the acceptance criteria for the journal 'Nanomaterials'. Specifically, I recommend publication if the authors address the following comments:
1. In the EDS analysis, the cobalt content of 0.51 atomic % may fall within the error range, suggesting potential unreliability in the result. I recommend that the authors employ additional verification methods, such as XPS.
2. The authors should present the gas response under various humidity conditions (RH 20%, RH 40%, RH 60%, and RH 80%) to assess water resistance.
3. There are several typographical errors in the English text, including terms like "zno" and "butanine". It would be beneficial to have the manuscript reviewed and refined by a professional editing service.
Except for minor typos in the manuscript, I believe it admirably meets the standards for acceptance. example "in line 38, zno should be ZnO" There are numerous errors regarding the subscript.
Reviewer 3 Report
The authors presented an interesting study on the effect of cobalt nanoparticles on the chemoresistive properties of zinc oxide in the detection of 2-butanone. However, several important aspects should be addressed before further publication:
1. authors are requested to seriously check the manuscript for a large number of formatting errors: lower indices, number and unit breaks, extra/necessary spaces, asterisks instead of multiplication signs, etc.
2. There is no need to capitalise n-type conductance.
3. it is not necessary to write degrees Celsius in full, there is a special term for this.
4. Figures are presented in very poor quality (e.g. Figures 2,6,8 are key to the paper, but little can be understood from them).
5. Row 149 - the degree designation is missing at the end of the row.
6. Unwarranted abbreviations are frequently used in the text, e.g. line 235 "temp".
7. The text needs to be proofread for a large number of typographical errors, e.g. line 269 "2-butanine".
8. Equation (6) is not explicitly equated.
9. What do the authors mean by: "UP water (18.25 MΩ*cm)". Clarification is needed.
10. The authors keep mentioning "overnight". Exactly how long is this?
11. The authors did not give the molar/atomic composition of the nanocomposite obtained. In what form is cobalt present in the nanocomposite? This is an important observation.
12. In line 117, the authors write "uniform thickness was formed". How was the uniform thickness controlled?
13. On line 151 the authors write: "In addition, there are no characteristic peaks of other elements in the XRD spectra of the four samples". It is very difficult to see specific elements in the XRD spectra, perhaps the authors meant phases?
14. Lines 151-154 are not repetitions of the same thing?
15. In line 185 the authors give the composition of the nanocomposite in mass fractions, I think it is more appropriate to give it in molar/atomic percent.
16. When discussing the high selectivity of the obtained composites towards 2-butanone, the authors refer to the large amount of beta-H. I would suggest that this designation should be disclosed according to the guidelines in the IUPAC names.
17. When discussing the detection mechanism and improved gas sensitivity, the authors state in line 309: "the high catalytic and oxygen adsorption capacity of cobalt particles". Is there a reference to support this statement?
18. Atmospheric humidity is not specified for gas sensitive experiments. Was it 0%? If so, what is the practical application of this study?
The quality of the English should be improved throughout the manuscript.
Round 2
Reviewer 1 Report
The authors has significantly improved the article and can be proceeded to the next publication process. However, the quality of all figures are still very poor. In addition, the XPS data for Co2p are incomplete and the XPS peaks for Co2p consists of significant satellite peaks that are unique for Co3O4 structures. The authors need to show clear and convincing XPS figure which explicitly supports their claim.
Reviewer 3 Report
The article in this form can now be recommended for publication.
The English in the article has been improved.
